# Updating Phospholipase A_2_ Biology

**DOI:** 10.3390/biom10101457

**Published:** 2020-10-19

**Authors:** Makoto Murakami, Hiroyasu Sato, Yoshitaka Taketomi

**Affiliations:** Laboratory of Microenvironmental and Metabolic Health Sciences, Center for Disease Biology and Integrative Medicine, Graduate School of Medicine, The University of Tokyo, Bunkyo-ku, Tokyo 113-8655, Japan; sato-hr@m.u-tokyo.ac.jp (H.S.); taketomiys@m.u-tokyo.ac.jp (Y.T.)

**Keywords:** fatty acid, knockout mouse, lipid mediator, lipidomics, lysophospholipid, membrane, phospholipase A_2_, phospholipid

## Abstract

The phospholipase A_2_ (PLA_2_) superfamily contains more than 50 enzymes in mammals that are subdivided into several distinct families on a structural and biochemical basis. In principle, PLA_2_ has the capacity to hydrolyze the *sn*-2 position of glycerophospholipids to release fatty acids and lysophospholipids, yet several enzymes in this superfamily catalyze other reactions rather than or in addition to the PLA_2_ reaction. PLA_2_ enzymes play crucial roles in not only the production of lipid mediators, but also membrane remodeling, bioenergetics, and body surface barrier, thereby participating in a number of biological events. Accordingly, disturbance of PLA_2_-regulated lipid metabolism is often associated with various diseases. This review updates the current state of understanding of the classification, enzymatic properties, and biological functions of various enzymes belonging to the PLA_2_ superfamily, focusing particularly on the novel roles of PLA_2_s in vivo.

## 1. Introduction

Based on their structural relationships, the PLA_2_ superfamily is classified into several families, including the secreted PLA_2_ (sPLA_2_), cytosolic PLA_2_ (cPLA_2_), Ca^2+^-independent PLA_2_ (iPLA_2_, also called patatin-like phospholipase (PNPLA)), platelet-activating factor acetylhydrolase (PAF-AH), lysosomal PLA_2_ (LPLA_2_), PLA/acyltransferase (PLAAT), α/β hydrolase (ABHD), and glycosylphosphatidylinositol (GPI)-specific PLA_2_ families. PLA_2_s trigger the production of lipid mediators by releasing polyunsaturated fatty acids (PUFAs) and lysophospholipids from membrane phospholipids, and also participate in membrane homeostasis by altering phospholipid composition, in energy production by supplying fatty acids for β-oxidation, in generation of barrier lipids, or in fine-tuning of the microenvironmental balance between saturated and unsaturated fatty acids, among others. Many of the PLA_2_ enzymes recognize the differences in the fatty acyl and/or head group moieties of their substrate phospholipids, and several enzymes catalyze even non-PLA_2_ reactions such as phospholipase A_1_ (PLA_1_), lysophospholipase, neutral lipid lipase, and transacylase reactions. The in vivo functions of individual PLA_2_s rely on their enzymatic, biochemical, and cell biological properties, their tissue and cellular distributions, lipid composition in target membranes, the spatiotemporal availability of downstream lipid-metabolizing enzymes, or the presence of cofactor(s) that can modulate the enzymatic function, in various pathophysiological settings.

During the last two decades, the functions of various PLA_2_s have been clarified by studies based on not only gene-manipulated (knockout and transgenic) mice but also human diseases caused by mutations of these enzymes. Here, we provide an overview of the biological roles of various PLA_2_s and their underlying lipid pathways, focusing mainly on new findings in the last five years. Readers interested in older views as a starting point for further readings should refer to our current reviews describing the classification of the PLA_2_ superfamily [1], those covering PLA_2_s and lipid mediators broadly [2,3], and those focusing on sPLA_2_s [4,5,6,7].

## 2. The sPLA_2_ Family

### 2.1. General Features

The sPLA_2_ family comprises low-molecular-mass, Ca^2+^-requiring enzymes with a conserved His-Asp catalytic dyad. There are 11 mammalian sPLA_2_s (IB, IIA, IIC, IID, IIE, IIF, III, V, X, XIIA, and XIIB), which are structurally subdivided into group I/II/V/X, group III, and group XII branches [8]. Individual sPLA_2_s exhibit unique tissue or cellular distributions and enzymatic properties, suggesting their distinct biological roles. With regard to the substrate specificity of sPLA_2_s as assessed by an assay using tissue-extracted natural membranes, sPLA_2_-IB, -IIA and -IIE do not discriminate *sn*-2 fatty acid species, sPLA_2_-V tends to prefer those with a lower degree of unsaturation such as oleic acid (OA; C18:1) and linoleic acid (LA; C18:2), and sPLA_2_-IID, -IIF, -III and -X tend to prefer PUFAs such as ω6 arachidonic acid (AA; C20:4) and ω3 docosahexaenoic acid (DHA; C22:6). With regard to the polar head groups, sPLA_2_-III, -V and -X efficiently hydrolyze phosphatidylcholine (PC), while sPLA_2_s in the group II subfamily hydrolyze phosphatidylethanolamine (PE) much better than PC. Individual sPLA_2_s exert their specific functions by producing lipid mediators, by altering membrane phospholipid composition, by degrading foreign phospholipids in microorganisms or diets, or by modifying extracellular non-cellular lipid components such as lipoproteins, pulmonary surfactant or microvesicles in response to given microenvironmental cues. In certain cases, the sPLA_2_-binding protein PLA2R1 modulates the functions of sPLA_2_s in either a positive or negative way. The pathophysiological roles of individual sPLA_2_s, as revealed by studies using sPLA_2_ knockout or transgenic mice in combination with comprehensive lipidomics, have been detailed in several recent reviews [3,6,7,9].

### 2.2. sPLA_2_-IB in Digestion and Immunity

sPLA_2_-IB (encoded by *PLA2G1B* in human) is synthesized as an inactive zymogen in the pancreas, and its *N*-terminal propeptide is cleaved by trypsin to yield an active enzyme in the duodenum [10]. The main role of sPLA_2_-IB, a “pancreatic sPLA_2_”, is to digest dietary and biliary phospholipids in the intestinal lumen. Perturbation of this process by gene disruption or pharmacological inhibition of sPLA_2_-IB leads to resistance to diet-induced obesity, insulin resistance, and atherosclerosis due to decreased phospholipid digestion and absorption in the gastrointestinal tract [11,12,13,14]. Indeed, the human *PLA2G1B* gene maps to an obesity susceptibility locus [15]. These functions of sPLA_2_-IB have been summarized in a recent review [16].

Beyond the well-established role of sPLA_2_-IB as a “digestive sPLA_2_” as outlined above, two recent studies have uncovered novel immunological functions of this sPLA_2_. Entwistle et al. [17] showed that sPLA_2_-IB is induced in a population of intestinal epithelial cells during helminth infection and is responsible for killing tissue-embedded larvae. *Pla2g1b*^−/−^ mice fail to expel the intestinal helminths *Heligmosomoides polygyrus* and *Nippostrongylus brasiliensis*. Treatment of the parasite with sPLA_2_-IB hydrolyzes worm phospholipids (e.g., PE) and impairs development to the adult stage, suggesting that exposure to this enzyme in the intestine is an important mechanism of host-mediated defense against such parasites.

Pothlichet et al. [18] reported that sPLA_2_-IB is involved in CD4^+^ T cell lymphopenia in patients infected with human immunodeficiency virus (HIV). sPLA_2_-IB, in synergy with the HIV gp41 envelope protein, induces CD4^+^ T cell anergy, inhibiting the responses to IL-2, IL-4, and IL-7 as well as activation, proliferation, and survival of CD4^+^ T cells. Other sPLA_2_s fail to display a similar function, implying a specific action of sPLA_2_-IB. Importantly, the effects of HIV on CD4^+^ T cell anergy can be blocked by a sPLA_2_-IB-specific neutralizing antibody in vivo. Thus, the sPLA_2_-IB/gp41 pair constitutes a new mechanism of immune dysfunction, although the cellular source of plasma sPLA_2_-IB in this context remains to be determined.

### 2.3. sPLA_2_-IIA in Host Defense, Sterile Inflammation, and Colon Cancer

The best-known physiological function of sPLA_2_-IIA (encoded by *PLA2G2A*) is the degradation of bacterial membranes, particularly those in Gram-positive bacteria, thereby providing the first line of antimicrobial defense as a “bactericidal sPLA_2_” [19,20]. The ability of sPLA_2_-IIA to hydrolyze PE and phosphatidylglycerol, which are abundant in bacterial membranes, appears to fit with its anti-bacterial function. In the lungs of patients with cystic fibrosis, sPLA_2_-IIA-resistant Gram-negative *Pseudomonas aeruginosa* upregulates the expression of sPLA_2_-IIA, which then eradicates sPLA_2_-IIA-sensitive Gram-positive *Staphylococcus aureus*, allowing the former bacterium to become dominant within the niche [21]. Thus, sPLA_2_-IIA-mediated regulation of the bacterial community in the lung microenvironment crucially affects the pathology of cystic fibrosis.

sPLA_2_-IIA is often referred to as an “inflammatory sPLA_2_”, as its expression is induced by pro-inflammatory cytokines and lipopolysaccharide (LPS) [22]. Besides its action on bacterial membranes as noted above, sPLA_2_-IIA targets phospholipids in extracellular microvesicles (EVs), particularly those in extracellular mitochondria (organelles that have evolved from bacteria), which are released from activated platelets or leukocytes at sites of inflammation [23]. Hydrolysis of EV phospholipids by sPLA_2_-IIA results in the production of lipid mediators as well as the release of mitochondrial DNA as a danger-associated molecular pattern (DAMP) that contributes to amplification of inflammation. Specifically, the AA released from platelet-derived EVs by sPLA_2_-IIA is metabolized by platelet-type 12-lipoxygenase to 12-hydroxyeicosatetraenoic acid (HETE), which then acts on the BLT2 receptor on neutrophils facilitating the uptake of the EVs [24]. Thus, sPLA_2_-IIA is primarily involved in host defense by killing bacteria and triggering innate immunity, while over-amplification of the response leads to exacerbation of inflammation by hydrolyzing EVs (Figure 1). These dual functions of sPLA_2_-IIA were summarized in a recent review [25].

sPLA_2_-IIA also appears to play a role in host defense against the malaria pathogen *Plasmodium falciparum*. Several sPLA_2_s, including sPLA_2_-IIF, -V and -X, which efficiently hydrolyze plasma lipoproteins to release free fatty acids, have the capacity to inhibit parasite growth in vitro, yet these sPLA_2_s are undetectable in human plasma. sPLA_2_-IIA, though hardly hydrolyzing normal lipoproteins, is increased in the plasma of malaria patients and hydrolyzes “oxidized” lipoproteins to block *Plasmodium* growth [26]. Injection of recombinant sPLA_2_-IIA into *Plasmodium*-infected mice reduces the peak of parasitemia when the level of plasma peroxidation is increased during infection. Thus, malaria-induced oxidation of lipoproteins converts them into a preferential substrate for sPLA_2_-IIA, thus promoting its parasite-killing effect.

*PLA2G2A*-transgenic mice display notable skin abnormalities with hair loss and epidermal hyperplasia [27]. Using *K14*-driven, skin-specific *PLA2G2A*-transgenic mice, Chovatiya et al. [28] recently demonstrated that sPLA_2_-IIA promotes hair follicle stem cell proliferation through JNK signaling and that sPLA_2_-IIA-knockdown skin cancers xenografted into NOD-SCID mice show a concomitant reduction of tumor volume and decreased JNK signaling. Kuefner et al. [29] showed that *PLA2G2A*-transgenic mice are protected from diet-induced obesity and become more prone to adipocyte browning with increased expression of thermogenic markers. However, since C57BL/6 mice do not express sPLA_2_-IIA endogenously due to a frameshift mutation in the *Pla2g2a* gene [30], the physiological relevance of these results obtained from transgenic overexpression of sPLA_2_-IIA in this mouse strain should be interpreted with caution. The increased energy expenditure in *PLA2G2A*-transgenic mice might be simply due to their lack of fur, which results in increased heat dissipation from the body surface. Indeed, transgenic mice overexpressing sPLA_2_-IIF or sPLA_2_-X are also furless and lean [31]. Studies using a series of sPLA_2_-knockout mice have suggested that sPLA_2_-IIF and sPLA_2_-IID are the main endogenous regulators of skin homeostasis and adipocyte browning, respectively, as described below.

Although sPLA_2_-IIA has been identified as a genetic modifier of mouse intestinal tumorigenesis [30], the underlying mechanism has long remained unclear. Schewe et al. [32] recently provided a potential mechanism by which sPLA_2_-IIA, expressed by Paneth cells in the small intestine, suppresses colon cancer. In a normal state, sPLA_2_-IIA inhibits Wnt signaling through intracellular activation of Yap1. Upon inflammation, sPLA_2_-IIA is secreted into the intestinal lumen, where it promotes inflammation through prostaglandin (PG) E_2_ synthesis via the PLA2R1-cPLA_2_α pathway and Wnt signaling. Transgenic overexpression of sPLA_2_-IIA delays recovery from colonic inflammation but decreases colon cancer susceptibility due to perturbation of its homeostatic Wnt-inhibitory function. Thus, this trade-off effect could provide a mechanism whereby sPLA_2_-IIA acts as a genetic modifier of colonic inflammation and cancer. However, the inflammatory sPLA_2_-IIA-PLA2R1-cPLA_2_α-PGE_2_ signaling axis proposed in this study may require further clarification, since contrary to this hypothesis, there is ample evidence that cPLA_2_α and its product PGE_2_ contribute to attenuation of colitis and promotion of colon cancer [33]. Other factors, such as intestinal dysbiosis due to loss of sPLA_2_-IIA, might primarily underlie the increased susceptibility to gastrointestinal cancer—a possibility that awaits future study.

Apart from its potential signaling role, PLA2R1 acts as a clearance receptor for sPLA_2_s [8,34]. In a model of experimental autoimmune myocarditis induced by immunizing BALB/c mice, a strain possessing a normal *Pla2g2a* gene, with the murine α-myosin heavy chain [30], PLA2R1 deficiency markedly increases sPLA_2_-IIA and -IB proteins (but not mRNAs) in the myocardium, probably as a result of their impaired clearance [35]. In the affected myocardium, PLA2R1 and these two sPLA_2_s are localized in α-SMA^+^ myofibroblasts and infiltrating neutrophils, respectively. The *Pla2r1*^−/−^ myocardium shows increased areas of inflammatory cell infiltration, accompanied by an increase in PGE_2_, which promotes IL-23-induced expansion of Th17 cells. Thus, it appears that this increase of sPLA_2_-IIA and -IB proteins due to their defective clearance contributes to exacerbation of autoimmune myocarditis, although it remains unclear whether these sPLA_2_s act through PGE_2_ synthesis or through other mechanisms, whether some other sPLA_2_s that are expressed in the myocardium and have the capacity to bind to PLA2R1 are also involved in this process, or whether the effect of PLA2R1 ablation is sPLA_2_-independent.

### 2.4. sPLA_2_-IID in Immunosuppression, Host Defense, and Adaptive Thermogenesis

sPLA_2_-IID (encoded by *PLA2G2D*), constitutively expressed in dendritic cells (DCs) in lymphoid organs, is a “resolving sPLA_2_” that attenuates DC-mediated adaptive immunity by hydrolyzing PE to mobilize anti-inflammatory ω3 PUFAs and their metabolites such as DHA-derived resolvin D1 [36]. *Pla2g2d*^−/−^ mice exhibit more severe contact hypersensitivity (Th1 response) and psoriasis (Th17 response), accompanied by marked reductions of ω3 PUFAs and their metabolites in the draining lymph nodes and spleen. On the other hand, *Pla2g2d*^−/−^ mice are protected against skin cancer likely because of the enhanced anti-tumor immunity in association with increased IFN-γ^+^CD8^+^ T cells [37]. The immunosuppressive role of sPLA_2_-IID in the Th1, Th2, and Th17 responses was detailed in a recent review [9].

Chronic low-grade inflammation is associated with age-related immune dysfunction in the lung, and is countered by enhanced expression of pro-resolving/anti-inflammatory factors to maintain tissue homeostasis. Vijay et al. [38] reported that sPLA_2_-IID with an anti-inflammatory property contributes to worse outcomes in mice infected with severe acute respiratory syndrome coronavirus (SARS-CoV) or influenza A virus. Strikingly, *Pla2g2d*^−/−^ mice are highly protected against SARS-CoV infection, with enhanced migration of DCs to the lymph nodes, augmented anti-viral T cell responses, reduced lung damage, and increased survival. In the context of the current worldwide SARS-CoV2 pandemic, inhibition of sPLA_2_-IID in the lungs of older patients with severe respiratory infections could be a potentially attractive therapeutic intervention for restoration of immune function.

ω3 PUFAs confer health benefits by preventing inflammation and obesity and by increasing thermogenesis in brown and beige adipocytes. sPLA_2_-IID is constitutively expressed in M2 macrophages in white adipose tissue (WAT) and downregulated during obesity [39]. Global or macrophage-specific sPLA_2_-IID deficiency decreases energy expenditure and thermogenesis by preventing adipocyte browning, thus exacerbating diet-induced obesity, insulin resistance, and WAT inflammation [39]. In WAT, PLA2G2D constitutively supplies a pool of ω3 PUFAs, which acts on the PUFA receptor GPR120 and thereby promotes the thermogenic program as a “thermogenic sPLA_2_”. Importantly, dietary supplementation with ω3 PUFAs normalizes the metabolic derangement in *Pla2g2d*^−/−^ mice. These findings highlight the contribution of the macrophage-driven PLA2G2D-ω3 PUFA axis to metabolic health (Figure 2). Possibly in relation to this, a polymorphism in the human *PLA2G2D* gene is linked to body weight changes in patients with chronic lung disease [40].

### 2.5. sPLA_2_-IIE and sPLA_2_-IIF in the Skin

sPLA_2_-IIE (encoded by *PLA2G2E*) is hardly detected in human tissues, whereas sPLA_2_-IIE instead of sPLA_2_-IIA is upregulated in several mouse tissues under inflammatory or other conditions. For instance, sPLA_2_-IIE is highly upregulated in adipocytes during diet-induced obesity [41], and is expressed in hair follicles in correlation with the growth phase of the hair cycle [42]. sPLA_2_-IIE hydrolyzes PE without apparent fatty acid selectivity in lipoproteins and hair follicles, and *Pla2g2e*^−/−^ mice display modest metabolic and hair follicle abnormalities.

sPLA_2_-IIF (encoded by *PLA2G2F*) is an “epidermal sPLA_2_” expressed in the suprabasal epidermis and upregulated by the Th17 cytokines IL-17A and IL-22 in psoriatic skin [31]. sPLA_2_-IIF preferentially hydrolyzes PUFA-containing plasmalogen-type PE to produce lysoplasmalogen (plasmalogen-type lysophosphatidylethanolamine; P-LPE), which in turn promotes epidermal hyperplasia. Accordingly, *Pla2g2f*^−−^ mice are protected against psoriasis and skin cancer, while *Pla2g2f*-transgenic mice spontaneously develop psoriasis-like skin and are more susceptible to skin cancer [31]. Overall, two skin sPLA_2_s, sPLA_2_-IIE in the outer root sheath of hair follicles and sPLA_2_-IIF in epidermal keratinocytes, play non-redundant roles in distinct compartments of mouse skin, underscoring the functional diversity of multiple sPLA_2_s in the coordinated regulation of skin homeostasis and diseases. The roles of sPLA_2_-IIE and -IIF were detailed in a recent review [9].

### 2.6. sPLA_2_-III in Male Reproduction, Anaphylaxis, and Colonic Diseases

sPLA_2_-III (encoded by *PLA2G3*) consists of three domains, in which the central sPLA_2_ domain similar to bee venom group III sPLA_2_ is flanked by large and unique *N*- and *C*-terminal domains [43]. The enzyme is processed to the sPLA_2_ domain-only form that retains full enzymatic activity [44]. sPLA_2_-III is expressed in the epididymal epithelium and acts on immature sperm cells passing through the epididymal duct in a paracrine manner to allow sperm membrane phospholipid remodeling, a process that is prerequisite for sperm motility [45]. Homozygous and even heterozygous *Pla2g3*-deficient sperm have impaired motility and thereby fail to fertilize oocytes, leading to hypofertility. In the context of allergy, sPLA_2_-III secreted from immature mast cells is functionally coupled with lipocalin-type PGD_2_ synthase (L-PGDS) in neighboring fibroblasts to supply a microenvironmental pool of PGD_2_, which in turn acts on the PGD_2_ receptor DP1 on mast cells to promote their proper maturation [46]. Accordingly, mice lacking sPLA_2_-III, as well as those lacking L-PGDS or DP1, have immature mast cells and display reduced local and systemic anaphylactic responses.

Several lines of evidence suggest a potential link between sPLA_2_-III and the development of colon cancer. For instance, sPLA_2_-III-transfected colon cancer cells xenografted into nude mice show increased growth [47], higher expression of sPLA_2_-III in human colorectal cancer is positively correlated with a higher rate of lymph node metastasis and shorter survival [48], and polymorphisms in the human *PLA2G3* gene are significantly associated with a higher risk of colorectal cancer [49]. Importantly, *Pla2g3*^−/−^ mice are resistant to several models of colon cancer [50]. Furthermore, *Pla2g3*^−/−^ mice are less susceptible to colitis, with lower expression of pro-inflammatory and pathogenic Th17 cytokines and higher expression of epithelial barrier genes [50], implying that amelioration of colonic inflammation by sPLA_2_-III ablation underlies the protective effect against colon cancer. The *Pla2g3*^−/−^ colon displays significant reduction of several lysophospholipids including lysophophatidic acid (LPA) and lysophosphatidylinositol (LPI) [50], which may promote colon inflammation or cancer through their receptors LPA_2_ and GPR55, respectively [51,52]. Overall, these results establish a role for sPLA_2_-III in the aggravation of colonic inflammation and cancer and point to sPLA_2_-III as a novel druggable target for colorectal diseases. The biological roles of sPLA_2_-III were summarized in a recent review [9].

### 2.7. sPLA_2_-V in Obesity, Type-2 Immunity, and Aortic Protection

Although sPLA_2_-V (encoded by *PLA2G5*) was previously thought to be a regulator of AA metabolism [53,54], it has become obvious that this sPLA_2_ has a preference for phospholipids bearing *sn*-2 fatty acids with a lower degree of unsaturation. Transgenic overexpression of sPLA_2_-V results in neonatal death due to a respiratory defect attributable to the ability of sPLA_2_-V to potently hydrolyze dipalmitoyl-PC, a major component of lung surfactant [55]. Mice that are transgenic for other sPLA_2_s do not exhibit such a phenotype, implying the particular ability of sPLA_2_-V to hydrolyze PC with *sn*-2 palmitic acid (C16:0) in the lung microenvironment. sPLA_2_-V is markedly induced in adipocytes during obesity as a “metabolic sPLA_2_” and hydrolyzes PC in hyperlipidemic LDL to release OA and LA, which counteract adipose tissue inflammation and thereby ameliorate metabolic disorders [41]. Impairment of this process in *Pla2g5*^−/−^ mice leads to exacerbation of diet-induced obesity and insulin intolerance, accompanied by elevated phospholipid levels in plasma LDL. This phenotype is reminiscent of clinical evidence that a *PLA2G5* polymorphism is associated with plasma LDL levels in patients with type 2 diabetes [56] and that the levels of *PLA2G5* mRNA expression in WAT are inversely correlated with plasma LDL levels in obese subjects [41].

sPLA_2_-V is a “Th2-prone sPLA_2_” induced in M2 macrophages by the Th2 cytokines IL-4 and IL-13 and promotes Th2-driven pathologies such as asthma. Gene ablation of sPLA_2_-V perturbs proper polarization and function of M2 macrophages in association with decreased Th2 immunity [57]. *Pla2g5*^−/−^ mice show reduced activation of type 2 innate lymphoid cells (ILC2) and infiltration of eosinophils in the lung following repetitive inhalation of the fungal allergen *Alternaria Alternata* [58]. Adoptive transfer experiments have revealed the contribution of sPLA_2_-V expressed in both macrophages and non-hematopoietic cells (probably bronchial epithelial cells) to the pathology. Lipidomics analysis has demonstrated reduction of OA and LA in the lung and macrophages in *Pla2g5*^−/−^ mice. Exogenous administration of these unsaturated fatty acids to *Pla2g5*^−/−^ mice restores IL-33-induced inflammation and ILC2 expansion, implying that macrophage-associated sPLA_2_-V contributes to type 2 immunity by promoting ILC2 activation though the release of OA and LA. The biological roles of sPLA_2_-V in asthma were summarized in a recent review [59]. Probably because of the alteration in the macrophage phenotype, *Pla2g5*^−/−^ macrophages have a reduced ability to phagocytose extracellular materials, thereby being more susceptible to fungal infection and arthritis due to defective clearance of hazardous fungi and immune complexes, respectively [60,61]. Likewise, *Pla2g5*^−/−^ mice suffer from more severe lung inflammation caused by bacterial infection [62], which could also be explained by poor clearance of these microbes by alveolar macrophages. Additionally, local generation of LPE in the plasma membrane by sPLA_2_-V may also contribute to macrophage phagocytosis [63].

Aortic dissection is a life-threatening aortopathy involving separation of the aortic wall. Since aortic dissection occurs suddenly without preceding clinical signs and current treatment strategies are limited mainly to antihypertensive agents and emergency surgery, biomarkers that can predict fragility and/or therapeutic targets for stabilization of the aortic wall are needed in order to improve patient outcomes. Behind its proposed role in atherosclerosis development [64], sPLA_2_-V is a primary “endothelial sPLA_2_” that protects against aortic dissection by endogenously mobilizing vasoprotective fatty acids [65]. Global and endothelial cell-specific deletion of sPLA_2_-V leads to dissection of the thoracic ascending aorta shortly after infusion of angiotensin II (AT-II). In the AT-II-treated aorta, endothelial sPLA_2_-V mobilizes OA and LA, which attenuate endoplasmic reticulum (ER) stress and increase the expression of lysyl oxidase, an enzyme that crosslinks extracellular matrix (ECM) proteins, thereby stabilizing the ECM in the aorta. Of note, dietary supplementation with OA and LA reverses the increased susceptibility of *Pla2g5*^−/−^ mice to aortic dissection. These findings reveal an unexplored functional link between sPLA_2_-driven phospholipid metabolism and aortic stability (Figure 3), possibly contributing to the development of improved diagnostic and/or therapeutic strategies for preventing aortic dissection. Importantly, this work provides in vivo relevance for the actions of this sPLA_2_ that had been proposed by several in vitro studies: (i) it releases OA and LA in preference to PUFAs, (ii) it preferentially acts on membranes of agonist-stimulated rather than quiescent cells, and (iii) it is retained on the cell surface through binding to heparan sulfate proteoglycan. Furthermore, this avenue of cardiovascular research has revealed a potential mechanism that could underlie the benefits of the olive oil-rich (i.e., OA-rich) Mediterranean diet in terms of cardiovascular health.

### 2.8. sPLA_2_-X in Sperm Activation, Colitis, and Asthma

Among the mammalian sPLA_2_s, sPLA_2_-X (encoded by *PLA2G10*) has the highest activity on PC leading to release of fatty acids, particularly PUFAs, and is activated by cleavage of the *N*-terminal propeptide by furin-type convertases [66]. In mice, sPLA_2_-X is expressed abundantly in the testis and gastrointestinal tract and to a much lesser extent in the lung, whereas its expression in other tissues is very low. In the process of reproduction, sPLA_2_-X is secreted from the acrosomes of activated spermatozoa and hydrolyzes sperm membrane phospholipids to release DHA, docosapentaenoic acid (DPA, C22:5), and LPC, which facilitate in vitro fertilization with oocytes [33,67]. *Pla2g10*^−/−^ spermatozoa also show impairment of the late phase of the progesterone-induced acrosome reaction ex vivo [68]. Thus, the two particular sPLA_2_s expressed in male reproductive organs, i.e., sPLA_2_-III secreted from the epididymal epithelium (see above) and sPLA_2_-X secreted from sperm acrosomes, act as “reproductive sPLA_2_s” to coordinately regulate male reproduction.

sPLA_2_-X is expressed abundantly in colorectal epithelial and goblet cells and plays a protective role against colitis by mobilizing anti-inflammatory ω3 PUFAs such as EPA and DHA [33]. Accordingly, *Pla2g10*^−/−^ mice display more severe epithelial damage and inflammation with reduction of colonic ω3 PUFAs rather than ω6 AA in a colitis model, while *PLA2G10*-transgenic mice exhibit global anti-inflammatory phenotypes in association with elevation of systemic levels of ω3 PUFAs and their metabolites [33]. Supplementation with exogenous EPA restores the colitis phenotype in *Pla2g10*^−/−^ mice. Furthermore, *Pla2g10*^−/−^ mice have lower fecal LA levels [69], suggesting that gastrointestinal sPLA_2_-X may have a role in the digestion of dietary and biliary phospholipids (as in the case of sPLA_2_-IB) or, alternatively, contribute to shaping of the gut microbiota. The latter possibility may help to explain the fact that *Pla2g10*^−/−^ mice display certain inflammatory, cardiovascular, and metabolic phenotypes that are not necessarily consistent among different laboratories [69,70,71,72].

sPLA_2_-X is expressed constitutively in the airway epithelium and increased after antigen challenge in mice, and also in asthma patients. *Pla2g10*^−/−^ mice are protected from antigen-induced asthma, with marked reductions of airway hyperresponsiveness, eosinophil and T cell trafficking to the airways, airway occlusion, secretion of type-2 cytokines, generation of antigen-specific IgE, and synthesis of pulmonary eicosanoids including cysteinyl leukotrienes [73]. Further, *Pla2g10*^−/−^ mice have reduced IL-33 levels and fewer ILC2 cells in the lung, lower IL-33-induced IL-13 expression in mast cells, and a marked reduction in both the number of newly recruited macrophages and the M2 polarization of these macrophages in the lung [74]. These results indicate that sPLA_2_-X serves as a key regulator of both innate and adaptive immune responses to allergens. Interestingly, as in the case of bee venom group III sPLA_2_ that elicits strong type-2 immune responses, exogenous administration of sPLA_2_-X serves as an adjuvant, leading to augmented type-2 immune responses with increased airway hypersensitivity and antigen-specific type-2 inflammation following peripheral sensitization and subsequent airway challenge with the antigen [75]. The biological roles of sPLA_2_-X in asthma were detailed in a recent review [76].

## 3. The sPLA_2_ Family

### 3.1. General Features

The cytosolic PLA_2_ (cPLA_2_) family comprises 6 isoforms (α–ζ), among which cPLA_2_β, δ, ε, and ζ map to the same chromosomal locus [77]. There is structural similarity between the cPLA_2_ and iPLA_2_ families in that their catalytic domain is characterized by a three-layer α/β/α architecture employing a conserved Ser/Asp catalytic dyad [78,79]. It appears that these two families were evolved from a common ancestral gene, with the cPLA_2_ family emerging from the iPLA_2_ family at the branching point of vertebrates in correlation with the development of the lipid mediator signaling pathways. Enzymes belonging to the cPLA_2_ family are characterized by the presence of a C2 domain at their *N*-terminal region, with the exception of cPLA_2_γ in which this domain is absent. The C2 domain is responsible for the Ca^2+^-dependent association with membranes. Herein, we will overview several recent topics for cPLA_2_α as well as potential functions of other cPLA_2_ isoforms.

### 3.2. New Insights into cPLA_2_α

cPLA_2_α (group IVA PLA_2_; encoded by *PLA2G4A*) is the best known PLA_2_ that plays a central role in stimulus-coupled AA metabolism. cPLA_2_α is the only PLA_2_ that shows a striking substrate specificity for phospholipids containing AA (and also those containing EPA, if this ω3 PUFA is present in cell membranes at substantial levels). Upon cell activation, cPLA_2_α translocates from the cytosol to the perinuclear (particularly Golgi) membranes in response to an increase in the µM range of cytosolic Ca^2+^ concentration, and is maximally activated by phosphorylation through mitogen-activated protein kinases and other kinases [80,81]. The phosphoinositide PIP_2_ modulates the subcellular localization and activation of cPLA_2_α [82]. The regulatory roles of cPLA_2_α in eicosanoid generation in various pathophysiological events, as revealed by biochemical analyses as well as by studies using *Pla2g4a*^−/−^ mice, have been well summarized in several elegant reviews [83,84].

By means of comprehensive lipidomics, Slatter et al. [85] showed that human platelets acutely increase mitochondrial energy generation following stimulation and that the substrates for this, including multiple fatty acids and oxidized species that support energy generation via β-oxidation, are exclusively provided by cPLA_2_α. This implies that cPLA_2_α is a central regulator of both lipid mediator generation and energy flux in human platelets and that acute phospholipid membrane remodeling is required to support energy demands during platelet activation.

Ceramide-1-phosphate (C1P), a sphingolipid-derived bioactive lipid, directly binds to and activates cPLA_2_α to stimulate the production of eicosanoids in vitro [86], but in vivo evidence for this event has been lacking. Recently, MacKnight et al. [87] addressed this issue by generating knockin mice in which endogenous cPLA_2_α is replaced with a mutant form having an ablated C1P-interaction site. In a skin wound healing model, wound maturation, rather than wound closure, is enhanced in the mutant cPLA_2_α-knockin mice compared to control mice. Primary dermal fibroblasts from the knockin mice show substantially increased collagen deposition and migration. The knockin mice also show an altered eicosanoid profile, with a reduction of PGE_2_ and TXB_2_ (a stable end-metabolite of TXA_2_) as well as an increase of HETE species, which enhances the migration and collagen deposition of dermal fibroblasts. This gain-of-function role for the mutant cPLA_2_α is associated with its relocalization to the cytoplasm and cytoplasmic vesicles. These findings clarify the key mechanisms by which wound healing is regulated by cPLA_2_α-C1P interaction in vivo and provide insight into the roles of cPLA_2_α and eicosanoids in orchestrating wound repair.

Chao et al. [88] reported that the C2 domain in cPLA_2_α interacts with the CARD domain in mitochondrial antiviral signaling protein (MAVS), boosting NF-κB-driven transcriptional programs that promote experimental autoimmune encephalomyelitis, a model of multiple sclerosis. cPLA_2_α recruitment to MAVS also disrupts MAVS-hexokinase 2 interactions, decreasing hexokinase activity and the production of lactate involved in the metabolic support of neurons. These findings define a novel role of cPLA_2_α in driving pro-inflammatory astrocyte activities in cooperation with MAVS through protein–protein interaction in the context of neuroinflammation. It remains unknown whether some cPLA_2_α-driven lipid mediators are involved in this process.

Oncogenic *PIK3CA* (encoding a PI3K isoform) results in an increase of AA and eicosanoids, thus promoting cell proliferation to beyond a cell-autonomous degree. Mechanistically, mutant PIK3CA drives a multimodal signaling network involving mTORC2-PKCζ-mediated activation of cPLA_2_α [89]. Notably, inhibition of cPLA_2_α acts synergistically with a fatty acid-free diet to restore immunogenicity and selectively reduce mutant PIK3CA-induced tumorigenicity. This reveals an important role for activated PI3K signaling in regulation of AA metabolism, highlighting a targetable metabolic vulnerability that depends largely on dietary fat restriction.

### 3.3. cPLA_2_β Fusion Protein in Carcinoma Cell Proliferation and Survival

cPLA_2_β (group IVB PLA_2_; encoded by *PLA2G4B*) displays PLA_1_, PLA_2_ and more potent lysophospholipase activities in vitro. A kinetic study has demonstrated that cPLA_2_β associates with a membrane surface that is rich in phosphoinositides when intracellular Ca^2+^ is low, whereas it moves to a cardiolipin-rich membrane such as the mitochondrial membrane when intracellular Ca^2+^ rises. Among three splice variants termed cPLA_2_β1, β2 and β3, only the β3 form is identified as an endogenous protein and is constitutively associated with mitochondrial and early endosomal membranes [90].

Cheng et al. [91] reported that *JMJD7-PLA2G4B,* a read-through fusion gene formed by splicing of the neighboring *JMJD7* (jumonji domain containing 7) and *PLA2G4B* genes, is expressed in human squamous cell carcinoma (HNSCC), as well as several other cancers. Ablation of *JMJD7-PLA2G4B*, but not *JMJD7* or *PLA2G4B* alone, significantly inhibits proliferation of SCC cells by promoting G1 arrest and increases starvation-induced cell death. These findings provide a novel insight into the oncogenic control of JMJD7-PLA2G4B in HNSCC cell proliferation and survival and suggest that this fusion protein may serve as an important therapeutic target and prognostic marker for HNSCC development and progression. cPLA_2_β has also been implicated in age-related changes in phospholipids and decreased energy metabolism in monocytes [92]. However, it remains unknown whether certain lipid metabolites generated by cPLA_2_β would be involved in these events.

### 3.4. cPLA_2_γ in Lipid Droplet Formation

Human cPLA_2_γ (group IVC PLA_2_; encoded by *PLA2G4C*), lacking the C2 domain characteristic of the cPLA_2_ family, is *C*-terminally farnesylated, is tightly associated with membranes, and possesses lysophospholipase and transacylase activities in addition to PLA_2_ activity [93]. Several lines of evidence suggest that cPLA_2_γ acts mainly as a CoA-independent transacylase, transferring a fatty acid from one phospholipid to the other phospholipid, in cells [94,95]. cPLA_2_γ is widely expressed in human tissues with a tendency for higher expression in the heart and skeletal muscle, whereas its expression in most mouse tissues is very low, making it difficult to address the in vivo roles of cPLA_2_γ using a knockout strategy. Exceptionally, mouse cPLA_2_γ is highly expressed in oocytes during the stage of germinal vesicle breakdown, when it dynamically relocates from the cortex to the nuclear envelope, suggesting its possible role in nuclear membrane remodeling in developing oocytes [96].

Lipid droplet (LD) accumulation in hepatocytes is a typical characteristic of steatosis. Hepatitis C virus (HCV) infection, one of the risk factors related to hepatic steatosis, induces LD accumulation in human hepatocytes. cPLA_2_γ has been identified as a host factor upregulated by HCV infection and involved in HCV replication, where it promotes LD biogenesis and HCV assembly [97,98]. cPLA_2_γ, through the domain around the amino acid residues 260–292, is tightly associated normally with ER membranes and relocated into LDs. Importantly, *PLA2G4C* knockdown hampers LD formation upon HCV stimulation, while *PLA2G4C* overexpression leads to LD formation in hepatocytes and enhances LD accumulation in the liver of mice fed a high-fat diet, suggesting its potential role in fatty liver disease.

### 3.5. cPLA_2_δ in Psoriasis

cPLA_2_δ (group IVD PLA_2_; encoded by *PLA2G4D*) was first identified as a keratinocyte-specific cPLA_2_ isoform that is induced during psoriasis and releases LA selectively [99]. However, subsequent studies showed that the PLA_2_ activity of cPLA_2_δ is much weaker than that of cPLA_2_α [77] and that its PLA_1_ activity is superior to its PLA_2_ activity [90]. Cheung et al. [100] recently demonstrated the expression of cPLA_2_δ in psoriatic mast cells, and found unexpectedly that its activity is extracellular. IFN-α-stimulated mast cells release exosomes, which transfer cytoplasmic cPLA_2_δ to neighboring Langerhans cells expressing CD1a, which present lipid antigens to T cells. Thus, the exosome-mediated transfer of cPLA_2_δ from mast cells to Langerhans cells leads to the generation of neolipid antigens and subsequent recognition by lipid-specific CD1a-reactive T cells, inducing the production of IL-22 and IL-17A. These data offer an alternative model of psoriasis pathogenesis in which lipid-specific CD1a-reactive T cells contribute to psoriatic inflammation, suggesting that cPLA_2_δ inhibition or CD1a blockade may have potential for treatment of psoriasis. However, given that CD1a is present in humans but not in mice, and that cPLA_2_δ is located mainly in epidermal keratinocytes, the regulatory roles of cPLA_2_δ in psoriatic skin require further exploration.

### 3.6. cPLA_2_ε as an N-Acyltransferase for N-Acylethanolamine Biosynthesis

*N*-acylethanolamines (NAEs) represent a group of endocannabinoid lipid mediators, including arachidonoylethanolamine (also known as anandamide; AEA), which acts on the endocannabinoid receptors CB1 or CB2, and palmitoyletanolamine (PEA) and oleoylethanolamine (OEA), which act on the nuclear receptor PPARα or through other mechanisms. The biosynthesis of NAEs, particularly PEA and OEA, occurs in two steps; transfer of *sn*-1 saturated or monounsaturated fatty acid of PC to the amino group of PE by *N*-acyltransferases to generate *N*-acyl-PE (NAPE), followed by hydrolysis mainly by NAPE-specific phospholipase D (NAPE-PLD) to give rise to NAEs [101]. There are two types of *N*-acyltransferase, i.e., Ca^2+^-dependent and -independent enzymes. Recently, it has been shown that cPLA_2_ε (group IVE PLA_2_; encoded by *PLA2G4E*) functions as a Ca^2+^-dependent *N*-acyltransferase [102]. In response to Ca^2+^ ionophore stimulation, HEK293 cells overexpressing cPLA_2_ε produce various NAPE and NAE species, accompanied by concomitant decreases in PE and PC and increases in LPE and LPC, indicating that cPLA_2_ε produces NAPEs by utilizing the diacyl- and plasmalogen types of PE as acyl acceptors and the diacyl types of PC and PE as acyl donors [103] (Figure 4A). The activity of cPLA_2_ε is markedly enhanced by the presence of phosphatidylserine (PS) or other anionic phospholipids, and cPLA_2_ε largely co-localizes with PS in the plasma membrane and organelles involved in the endocytic pathway [104,105]. This localization might be related to the observation that cPLA_2_ε drives recycling through the clathrin-independent endocytic route [106].

However, it still remains to be determined whether cPLA_2_ε indeed contributes to NAPE and NAE biosynthesis under certain in vivo conditions. Since genetic deletion or pharmacological inhibition of NAPE-PLD disturbs lipid metabolism in the liver, intestine, and adipose tissue [107], as well as emotional behavior [108], it is tempting to speculate that cPLA_2_ε, which lies upstream of NAPE-PLD for NAE biosynthesis, may also participate in these events to improve metabolism and neuronal functions. The potential neuroprotective role of cPLA_2_ε is supported by the association between reduced expression of cPLA_2_ε and dementia, where adenoviral overexpression of cPLA_2_ε in hippocampal neurons completely restores cognitive deficits in the elderly APP/PS1 mouse, a model of Alzheimer’s disease [109].

### 3.7. cPLA_2_ζ in Myocardial Mitochondria

The observation that multiple fatty acids are non-selectively released from Ca^2+^-stimulated *Pla2g4a*^−/−^ lung fibroblasts, an event that is suppressed by cPLA_2_α inhibitors, suggests that other cPLA_2_ isoform(s) might contribute to this event. This fatty acid release and PGE_2_ production by *Pla2g4a*^−/−^ fibroblasts depend on cPLA_2_ζ (group IVF PLA_2_, encoded by *PLA2G4F*) [110]. In response to ionomycin, cPLA_2_ζ translocates to ruffles and dynamic vesicular structures, while cPLA_2_α translocates to the Golgi and ER, suggesting distinct mechanisms of regulation for the two enzymes.

Moon et al. [111] reported that mitochondria isolated from human heart contain at least two PLA_2_s—cPLA_2_ζ and iPLA_2_γ—of which cPLA_2_ζ mediates Ca^2+^-activated release of AA from mitochondria in normal heart. The AA pool mobilized by cPLA_2_ζ is preferentially channeled into cytochrome P450 epoxygenases for the synthesis of epoxyeicosatrienoic acids (EETs), which have a protective effect against heart failure. In contrast, in the failing heart, iPLA_2_γ mainly mobilizes mitochondrial AA, which preferentially couples with lipoxygenases for the synthesis of toxic HETEs that open mitochondrial permeability transition pores, leading to further progression of heart failure. These results reveal an unexplored biological role of cPLA_2_ζ as well as iPLA_2_γ (see below) in the myocardium, although confirmation using *Pla2g4f*-null mice will be required.

## 4. The iPLA_2_/PNPLA Family

### 4.1. General Features

The human genome encodes 9 Ca^2+^-independent PLA_2_ (iPLA_2_) enzymes. These enzymes are now more generally known as patatin-like phospholipase domain-containing lipases (PNPLA1-9), since all members in this family share a patatin domain, which was initially discovered in patatin (iPLA_2_α), a potato protein [2,112]. Unlike the cPLA_2_ family, which is present only in vertebrates, the iPLA_2_/PNPLA family is widely expressed in many eukaryotes including yeast, ameba, nematode, fly and vertebrates. iPLA_2_/PNPLA isoforms display lipid hydrolase or transacylase/acyltransferase activities with specificities for diverse lipids such as phospholipids, neutral lipids, sphingolipids, and retinol esters. In principle, enzymes with a large and unique *N*-terminal region (PNPLA6~9) act mainly on phospholipids as PLA_1_, PLA_2_ or lysophospholipases, whereas those lacking the *N*-terminal domain (PNPLA1~5) act on neutral lipids as lipases or transacylases. Analysis of gene disruption or mutation of the iPLA_2_/PNPLA enzymes in mice and humans have provided valuable insights into their physiological roles in homeostatic lipid metabolism that are fundamental for life. Since there have been many excellent reviews on this enzyme family [2,3,112,113], we will herein highlight several recent topics that shed further light on the pathophysiological roles of this enzyme family.

### 4.2. iPLA_2_β in Neurodegeneration and Hepatic Steatosis

iPLA_2_β (also known as group VIA PLA_2_; encoded by *PLA2G6*) is the only iPLA_2_ isoform that acts primarily as a PLA_2_ with poor fatty acid selectivity [114,115]. Different splice variants of iPLA_2_β are associated with the plasma membrane, mitochondria, ER, and the nuclear envelope. Although iPLA_2_β lacks a transmembrane domain, it has putative protein-interaction motifs such as ankyrin repeats, which are capable of interacting with multiple cognate receptor proteins, and a calmodulin-binding site, which interacts with the inhibitory calmodulin. The crystal structure of iPLA_2_β reveals a dimer formation of the catalytic domains, which are surrounded by ankyrin repeats that adopt an outwardly flared orientation, poised to interact with membrane proteins [116]. The closely integrated active sites are positioned for cooperative activation and internal transacylation. The structure also suggests allosteric inhibition by calmodulin, where a single calmodulin molecule interacts with two catalytic domains, altering the conformation of the dimer interface and active sites.

The roles of iPLA_2_β in male fertility, neuronal disorders, metabolic diseases, and inflammation, among others, have been studied in a number of *Pla2g6* knockout, knockdown, and overexpression studies and are well summarized in recent reviews [113,117]. In particular, the roles of iPLA_2_β in neuronal function have received extensive attention, since numerous mutations of the *PLA2G6* gene have been discovered in patients with neurodegenerative disorders such as infantile neuroaxonal dystrophy (INAD) and Parkinson’s disease [118]. In fact, iPLA_2_β is also referred to as the parkinsonism-associated protein PARK14, mutations of which are associated with impaired Ca^2+^ signaling in dopaminergic neurons [119].

Three recent studies using mutant flies have provided novel insights into the regulatory roles of iPLA_2_β in the brain in the context of Parkinson’s disease with α-synucleinopathy. Kinghorn et al. [120] reported that knockout of the *PLA2G6* gene in *Drosophila* results in reduced survival, locomotor deficits, and organismal hypersensitivity to oxidative stress, accompanied by mitochondrial abnormalities and increased lipid peroxidation levels. Inhibition of lipid peroxidation partially rescues the locomotor abnormalities and mitochondrial membrane potential caused by iPLA_2_β deficiency. Lin et al. [121] showed that the loss of iPLA_2_β causes an increase of brain ceramide, leading to lysosomal stress and neurodegeneration. iPLA_2_β binds to the retromer subunits Vps35 and Vps26 and enhances their function to promote protein and lipid recycling. Loss of iPLA_2_β impairs retromer function resulting in a progressive increase of ceramide, thus inducing a positive feedback loop that affects membrane fluidity and impairs retromer function and neuronal function. Mori et al. [122] showed that iPLA_2_β deficiency in *Drosophila* results in defective neurotransmission during the early developmental stages and progressive cell loss throughout the brain, including degeneration of the dopaminergic neurons. In the brain, iPLA_2_β loss results in shortening of the acyl-chain length of phospholipids, resulting in membrane lipid disequilibrium and thereby ER stress. Introduction of the mitochondria-ER contact site-resident protein C19orf12, another causal gene for Parkinson’s disease, in *PLA2G6*-deficient flies rescues the phenotypes associated with altered lipid composition, ER stress, and neurodegeneration. Moreover, the acceleration of α-synuclein aggregation by iPLA_2_β deficiency is suppressed by administration of LA, which corrects the brain phospholipid composition. Thus, membrane remodeling by iPLA_2_β is required for the survival of dopaminergic neurons and α-synuclein stability.

The roles of iPLA_2_β in phospholipid remodeling in the context of hepatic lipid metabolism have recently been studied using *Pla2g6*^−/−^ mice. iPLA_2_β deficiency attenuates obesity and hepatic steatosis in *ob/ob* mice through hepatic fatty-acyl phospholipid remodeling [123]. Aging sensitized by iPLA_2_β deficiency induces liver fibrosis and intestinal atrophy involving suppression of homeostatic genes and alteration of intestinal lipids and bile acids [124]. *Pla2g6*^−/−^ mice fed a high-fat diet show attenuation of hepatic steatosis through correction of defective phospholipid remodeling [125]. Lastly, *Pla2g6*^−/−^ mice fed a methionine/choline-deficient diet do not show correction of this defect, but the hepatocellular injury is attenuated via inhibition of lipid uptake genes [126].

### 4.3. iPLA_2_γ in Bioenergetics and Signaling

iPLA_2_γ (also known as group VIB PLA_2_, encoded by *PNPLA8*) displays PLA_2_ activity, but acts as a PLA_1_ toward phospholipids bearing *sn*-2 PUFA [127,128]. Accordingly, hydrolysis of PUFA-bearing phospholipids by iPLA_2_γ typically gives rise to 2-lysophospholipids (having a PUFA at the *sn*-2 position) rather than 1-lysophospholipids (having a saturated or monounsaturated acid at the *sn*-1 position). *Pnpla8*^−/−^ mice display multiple bioenergetic and neuronal dysfunctions, including growth retardation, kyphosis, muscle weakness with atrophy of myofilaments, cold intolerance, reduced exercise endurance, increased mortality due to cardiac stress, resistance to diet-induced obesity and insulin resistance, performance deficits in spatial learning and memory, and abnormal mitochondrial function with a dramatic decrease in fatty acid β-oxidation and oxygen consumption [129]. The alterations in myocardial and hippocampal cardiolipin content and composition indicate that iPLA_2_γ is involved in cardiolipin remodeling. These features of *Pnpla8*^−/−^ mice are reminiscent of those seen in patients with Barth syndrome, a disease caused by mutations in the human *TAZ* gene, which encodes tafazzin, a mitochondrial transacylase required for cardiolipin remodeling [130]. Moreover, loss-of-function variants of the human *PNPLA8* gene recapitulate the mitochondriopathy observed in *Pnpla8*^−/−^ mice [131].

iPLA_2_γ also participates in the generation of lipid mediators under certain conditions. *Pnpla8*^−/−^ mice display reduced ADP- or collagen-stimulated TXA_2_ generation by platelets ex vivo and show prolonged bleeding time and reduced pulmonary thromboembolism in vivo [132]. Cardiomyocyte-specific deletion of iPLA_2_γ decreases infarct size upon ischemia/reperfusion, with reduction of oxygenated metabolites of ω3 and ω6 PUFAs including PGs, HETEs, and hydroxy-DHAs [133]. iPLA_2_γ releases 9-hydroxyoctadecenoic acid (9-HODE), an oxygenated metabolite of LA, from cardiolipin, integrating mitochondrial bioenergetics and signaling [134]. Heart failure-induced activation of iPLA_2_γ leads to mitochondrial generation of HETEs that open the mitochondrial permeability transition pore, thus further amplifying myocardial damage [111]. Phospholipids bearing *sn*-2 AA are cleaved by the PLA_1_ activity of iPLA_2_γ to give rise to 2-arachidonoyl-lysophospholipids, which are then oxygenized directly by cyclooxygenase-2 or 12-lipoxygenase for conversion to eicosanoid-esterified lysophospholipids [134,135]. The generation of eicosanoid-esterified lysophospholipids is attenuated by the absence of iPLA_2_γ, underscoring an iPLA_2_γ-initiated pathway generating new classes of lipid metabolites with potential signaling functions.

### 4.4. PNPNA6 and PNPLA7 as Lysophospholipases

PNPLA6 (iPLA_2_δ) and its closest paralog PNPLA7 (iPLA_2_θ) have a lysophospholipase activity that cleaves LPC to yield fatty acid and glycerophosphocholine (GPC) [136,137]. Counterparts of these enzymes in yeast and fly act as a phospholipase B, which converts PC to GPC by liberating both *sn*-1 and *sn*-2 fatty acids. PNPLA6, also referred to as neuropathy target esterase (NTE), was originally identified as a target enzyme for the poisonous effect of organophosphates, which cause a severe neurological disorder characterized by degeneration of long axons in the spinal cord and peripheral nerves, leading to paralysis of the lower limbs [138]. In cultured renal cells, the production of GPC, an osmoprotective metabolite, is enhanced by PNPLA6 overexpression and is diminished by its siRNA knockdown or its inhibitor organophosphate [139]. Global *Pnpla6*^−/−^ mice die in utero due to placental defects, while neuron-specific *Pnpla6*^−/−^ mice exhibit progressive neuronal degeneration, leading to prominent neuronal pathology in the hippocampus and thalamus and also defects in the cerebellum [140,141]. Neuronal absence of PNPLA6 results in disruption of the ER, vacuolation of nerve cell bodies and abnormal reticular aggregates, and sustained elevation of PC over many months is accompanied by progressive degeneration and massive swelling of axons in the sensory and motor spinal tracts and worsening hindlimb dysfunction [142]. In humans, *PNPLA6* mutations near the catalytic site cause a severe motor neuron disease characterized by progressive spastic paraplegia and distal muscle wasting [143] as well as childhood blindness with retinal degeneration, including Leber congenital amaurosis, Oliver McFarlane syndrome, and Boucher-Neuhäuser syndrome [144]. Although the in vivo role of PNPLA7, also known as NTE-related esterase (NRE), is still unknown, it is downregulated by insulin in WAT [136] and interacts with LDs through its catalytic domain [145], suggesting its metabolic role.

### 4.5. PNPLA2 and PNPLA3 in Triglyceride Metabolism

Although PNPLA2 and PNPLA3 correspond to iPLA_2_ζ and iPLA_2_ε, respectively, according to the PLA_2_ classification, the latter names are not used here because these enzymes act essentially as neutral lipid lipases, but not as phospholipases. PNPLA2 is upregulated, while PNPLA3 is downregulated, upon starvation, and vice versa upon feeding, indicating that these two closely related lipases are nutritionally regulated in a reciprocal way [146]. PNPLA2, more generally known as adipose triglyceride lipase (ATGL), is a major lipase that hydrolyzes triglycerides in LDs to release fatty acids for β-oxidation-coupled energy production, a process known as lipolysis [147]. Genetic deletion or mutation of PNPLA2 leads to accumulation of triglycerides in multiple tissues leading to heart failure, while conferring protection from fatty liver and glucose intolerance, likely because these mice are able to utilize glucose but not free fatty acids as a fuel [148,149]. PNPLA2 deficiency also protects against cancer-associated cachexia by preventing fat loss [150]. The fatty acids released from LDs by PNPLA2 act as endogenous ligands for the nuclear receptor PPARα or PPARδ, which drives energy consumption [151,152]. In addition, the AA released from triglycerides by PNPLA2-driven lipolysis is utilized for eicosanoid generation in certain situations [153]. The activity of PNPLA2 is regulated positively by ABHD5 (also known as CGI-58) and negatively by perilipin, G0S2 and mysterin, which modulate the accessibility of PNPLA2 to LDs [152,154]. Adipocyte-specific *G0S2*-transgenic mice show attenuated lipolysis and adipocyte hypertrophy, accompanied by a reduced hepatic triglyceride level and increased insulin sensitivity [155], thus recapitulating the phenotypes observed in *Pnpla2*^−/−^ mice [149]. Mutations in the human *PNPLA2* gene cause Chanarin–Dorfman syndrome, a condition in which triglycerides are stored abnormally in the body [156]. *ABHD5* mutations also cause a similar neutral lipid storage disease but also additionally cause ichthyosis [157], likely because ABHD5 acts as a cofactor for not only PNPLA2-mediated lipolysis, but also PNPLA1-driven ω-*O*-acylceramide synthesis in the skin (see below). The regulatory mechanisms and metabolic roles of PNPLA2 have been detailed in other elegant reviews [158,159].

Mutations in the human *PNPLA3* gene are highly associated with non-alcoholic fatty liver disease (NAFLD) and steatohepatitis (NASH) [160]. However, *Pnpla3*^−/−^ mice do not display a fatty liver phenotype [161], whereas PNPLA3^I158M^ knockin mice develop hepatic steatosis [162], illustrating the complexity of the regulatory roles of PNPLA3 in hepatic lipid metabolism. PNPLA3 reportedly acts as an acyltransferase that alters the fatty acid composition of triglycerides [163], as a transacylase that promotes the transfer of PUFAs from triglycerides to phospholipids in hepatic LDs [164], or as a retinyl-palmitate lipase in hepatic stellate cells to fine-tune the plasma levels of retinoids, which might influence the differentiation of stellate cells to myofibroblasts [165]. It has now become recognized that PNPLA3 functions primarily as a triglyceride lipase and that I158M mutation leads to loss of function. Importantly, PNPLA3, and PNPLA3^I158M^ to an even greater extent, strongly interact with ABHD5, a cofactor for PNPLA2, thereby interfering with the lipolytic activity of PNPLA2 [166,167]. Overexpression of PNPLA3^I158M^ greatly suppresses PNPLA2-dependent lipolysis, leading to massive triglyceride accumulation in hepatocytes and brown adipocytes. Moreover, transgenic overexpression of PNPLA3^I158M^ increases hepatic triglyceride levels in WT mice, but not in *Abhd5*^−/−^ mice, confirming that the pro-steatotic effects of PNPLA3 require the presence of ABHD5. Thus, the increased abundance of PNPLA3^I148M^ results in sequestration of ABHD5 on LDs, thereby limiting the availability of this cofactor for activation of PNPLA2. The question remains of how PNPLA2, PNPLA3, and ABHD5 each find their way to hepatocyte LDs in the correct stoichiometric proportions and function to regulate LD assembly and turnover under conditions of fasting or nutrient excess.

### 4.6. PNPLA1 in Acylceramide Synthesis for Skin Barrier Function

ω-*O*-acylceramide, a unique sphingolipid present specifically in the *stratum corneum* of the epidermis, is essential for skin barrier formation, and impairment of its biosynthesis leads to ichthyosis or atopic dermatitis. Mutations in the human *PNPLA1* gene cause autosomal recessive congenital ichthyosis [168]. Unlike most PNPLA isoforms that are ubiquitously expressed in many tissues, PNPLA1 is localized predominantly in the upper layer of the epidermis. PNPLA1 acts as a unique transacylase, catalyzing the transfer of LA in triglyceride to the ω-hydroxy group of ultra-long-chain fatty acid in ceramide to give rise to ω-*O*-acylceramide [169,170,171] (Figure 4B). Global or keratinocyte-specific deletion of PNPLA1 hampers epidermal ω-*O*-acylceramide formation, thereby severely impairing skin barrier function leading to neonatal death due to excessive dehydration. The enzymatic activity of PNPLA1 is enhanced by ABHD5, which may present triglycerides to PNPLA1 to facilitate substrate recognition [172,173], providing a mechanism whereby *ABHD5* mutations cause Chanarin-Dorfman syndrome accompanied by ichthyosis with impaired ω-*O*-acylceramide formation. The role of PNPLA1 in ω-*O*-acylceramide biosynthesis has been described in detail in a recent review [174].

## 5. Other PLA_2_ Family

### 5.1. The PAF-AH Family

The PAF-acetylhydrolase (PAF-AH) family comprises one extracellular and three intracellular enzymes that were originally found to have the capacity to deacetylate and thereby inactivate the lysophospholipid-derived lipid mediator PAF [175,176]. Plasma-type PAF-AH (group VIIA PLA_2_; encoded by *PLA2G7*) is a secreted protein produced by macrophages, mast cells or other sources, and is now more generally referred to as lipoprotein-associated PLA_2_ (Lp-PLA_2_), existing as a low-density lipoprotein (LDL)-bound form in human plasma [177]. Although PAF is a potent mediator of allergic responses, Lp-PLA_2_ deficiency fails to augment airway inflammation or hyperresponsiveness after PAF+LPS treatment or passive or active allergic sensitization and challenge [178]. A series of clinical studies have revealed a correlation of Lp-PLA_2_ with atherosclerosis, likely because this enzyme hydrolyzes oxidized phospholipids (i.e., phospholipids having an oxidized fatty acid at the *sn*-2 position) in modified LDL with pro-atherogenic potential [179,180]. Although darapladib, a potent Lp-PLA_2_ inhibitor, failed to meet the primary endpoints of two large phase III trials for treatment of atherosclerosis [181], recent clinical and preclinical studies have revealed that Lp-PLA_2_ inhibition may have therapeutic effects in diabetic macular edema and Alzheimer’s disease [182,183]. Lp-PLA_2_ deficiency in *Apc^Min/+^* mice leads to decreased intestinal polyposis and tumorigenesis, suggesting a role of PAF or some oxidized lipids in cancer development [184].

Type-I PAF-AH is a heterotrimer composed of two catalytic α1 and α2 subunits (group XIIIA and XIIIB PLA_2_s, encoded by *PAFAH1B2* and *PAFAH1B3*, respectively), and a regulatory β subunit that is identical to LIS-1, a causative gene for a type of Miller-Dieker syndrome [185]. Loss of both the α1 and α2 catalytic subunits leads to male infertility [186], reduction of amyloid-β generation by promoting the degradation of amyloid precursor protein C-terminal fragments [187], and an increase in the size of the ganglionic eminences resulting from increased proliferation of GABAergic neurons through perturbation of the Wnt signaling pathway [188].

Type-II PAF-AH (PAF-AH2 or group VIIB PLA_2_; encoded by *PAFAH2*) shows significant homology with Lp-PLA_2_ and preferentially hydrolyzes oxidized phospholipids in cells. In a CCl_4_-induced liver injury model, *Pafah2*^−/−^ mice show a delay in hepatic injury recovery with unusual accumulation of 8-isoprostaglandin F_2α_ (8-iso-PGF_2α_) in membrane phospholipids, indicating that PAF-AH2 removes toxic oxidized lipids from cell membranes and thereby protects the tissue from oxidative stress-induced injury [189]. Mast cells spontaneously produce ω3 PUFA-derived epoxides (ω3 epoxides), such as EPA-derived 17,18-epoxyeicosatetraenoic acid (17,18-EpETE) and DHA-derived 19,20-epoxydocosapentaenoic acid (19,20-EpDPE), whose production depends on PAF-AH2-driven cleavage of ω3 epoxide-esterified phospholipids in mast cell membranes [190]. Genetic or pharmacological inactivation of PAF-AH2 reduces the steady-state production of ω3 epoxides, leading to attenuated mast cell activation and anaphylaxis following FcεRI crosslinking (Figure 5). Mechanistically, the ω3 epoxides promote IgE-mediated activation of mast cells by down-regulating Srcin1, a Src-inhibitory protein that counteracts FcεRI signaling, through a pathway involving PPARγ. Thus, the PAF-AH2–ω3 epoxide pathway ensures optimal mast cell activation and presents new potential drug targets for allergic diseases. The properties and functions of the PAF-AH family have been summarized in a recent review [191].

### 5.2. Lysosomal PLA_2_s

Lysosomal PLA_2_ (LPLA_2_ or group XV PLA_2;_ encoded by *PLA2G15*) is catalytically active under mildly acidic conditions and structurally homologous with lecithin:cholesterol acyltransferase (LCAT), an enzyme that transfers the *sn*-2 fatty acid of PC to cholesterol to produce cholesteryl ester in high-density lipoprotein (HDL) [192]. LPLA_2_ hydrolyzes both *sn*-1 and *sn*-2 fatty acids in phospholipids and contributes to phospholipid degradation in lysosomes. Genetic deletion of LPLA_2_ results in unusual accumulation of non-degraded lung surfactant phospholipids in lysosomes of alveolar macrophages leading to phospholipidosis [193], reduced presentation of lysophospholipid antigens to CD1d by invariant natural killer T (iNKT) cells [194], and impairment of adaptive T cell immunity against mycobacterium [195].

Peroxiredoxin 6 (Prdx6) is another lysosomal PLA_2_ that is also called acidic Ca^2+^-independent PLA_2_ (aiPLA_2_). Prdx6 is a multifunctional enzyme since it also possesses glutathione peroxidase and lysophospholipid acyltransferase activities. The aiPLA_2_ activity of Prdx6 has important physiological roles in the turnover (synthesis and degradation) of lung surfactant phospholipids [196], protection against LPS-induced lung injury [197], repair of peroxidized cell membranes [198], and sperm fertilizing competence [199]. The properties and functions of LPLA_2_ and Prdx6/aiPLA_2_ have been summarized in recent reviews [200,201].

### 5.3. The PLAAT Family

The PLAAT family, which comprises 3 enzymes in mice and 5 enzymes in humans, is structurally similar to lecithin:retinol acyltransferase (LRAT). Because of the ability of some members in this family to suppress *H-ras* signaling, it is also referred to as the HRASLS (for *H-ras*-like suppressor) family. Members of this family, including PLA2G16 (group XVI PLA_2_, also known as PLAAT3 or HRASLS3), display PLA_1_ and PLA_2_ activities, as well as Ca^2+^-independent *N*-acyltransferase activity that synthesizes NAPE, to various degrees [202]. PLA2G16 is highly expressed in adipocytes, and *Pla2g16*^−/−^ mice are resistant to diet-induced obesity [203]. Adipocyte-derived LPC produced by PLA2G16 activates NLRP3 inflammasomes in adipocytes and adipose tissue macrophages mediating homocysteine-induced insulin resistance [204]. PLA2G16 and its paralogs in this family have also been implicated in tumor invasion and metastasis [205], vitamin A metabolism [206], peroxisome biogenesis [207], and cellular entry and clearance of Picornaviruses [208]. The properties and functions of the PLAAT/PLA2G16 family have been summarized in recent reviews [209,210].

### 5.4. The ABHD Family

The ABHD family is a newly recognized group of lipolytic enzymes, comprising at least 19 enzymes in humans [211]. Enzymes in this family typically possess both hydrolase and acyltransferase motifs. Although the functions of many of the ABHD isoforms still remain uncertain, some of them have been demonstrated to act on neutral lipids or phospholipids as lipid hydrolases. ABHD3 selectively hydrolyzes phospholipids with medium-chain fatty acids [212]. ABHD4 releases fatty acids from multiple classes of *N*-acyl-phospholipids to produce *N*-acyl-lysophospholipids [213]. ABHD6 acts as lysophospholipase or monoacylglycerol lipase, the latter being possibly related to signaling of 2-arachidonoyl glycerol, an endocannabinoid lipid mediator that plays a role in retrograde neurotransmission [214]. ABHD12 hydrolyzes lysophosphatidylserine (LysoPS), and is therefore referred to as LysoPS lipase [215]. Genetic or pharmacological blockade of ABHD12 stimulates immune responses in vivo, pointing to a key role for this enzyme in regulating immunostimulatory lipid pathways. Mutations in the human *ABHD12* gene result in accumulation of LysoPS in the brain and cause a disease known as PHARC, which is characterized by polyneuropathy, hearing loss, ataxia, retinitis pigmentosa, and cataract [216]. ABHD16A acts as a PS-selective PLA_2_ (referred to as PS lipase), lying upstream of ABHD12 in the PS-catabolic pathway [217]. Disruption of ABHD12 and ABHD16A in macrophages respectively increases and decreases LysoPS levels and cytokine production. Although ABHD5 does not have catalytic activity because of the absence of a serine residue in the catalytic center, it interacts with and modifies the functions of several PNPLA members including PNPLA1-3, as described above.

### 5.5. GPI-Specific PLA_2_s

GPI is a complex glycolipid covalently linked to the C terminus of proteins on the plasma membrane, particularly in the raft microdomain. The biosynthesis of GPI and its attachment to proteins occur in the ER. GPI-anchoring proteins (GPI-APs) are subjected to fatty acid remodeling, which replaces an unsaturated fatty acid at the *sn*-2 position of the PI moiety with a saturated fatty acid. PGAP3, which resides in the Golgi, and PGAP6, which is localized mainly on the cell surface, are GPI-specific PLA_2_s (GPI-PLA_2_s) involved in fatty acid remodeling of GPI-APs [218]. PGAP3-dependent fatty acid remodeling of GPI-APs has a significant role in the control of autoimmunity, possibly by the regulation of apoptotic cell clearance and Th1/Th2 balance [219]. CRIPTO, a GPI-AP that plays critical roles in early embryonic development by acting as a Nodal co-receptor, is a highly sensitive substrate of PGAP6. CRIPTO is released by PGAP6 as a glycosyllysophosphatidylinositol-bound form and acts as a co-receptor in Nodal signaling. *Pgap6*^−/−^ mice show defects in early embryonic development, particularly in the formation of the anterior-posterior axis, as do *Cripto*^−/−^ embryos, suggesting that PGAP6 plays a critical role in Nodal signaling modulation through CRIPTO shedding [220].

## 6. Conclusions

In this review, we have provided an overview of the biological functions of a nearly full set of PLA_2_s identified to date, particularly over the past five years during which considerable advances in this research field have been made. In the interests of brevity, we have referenced previous reviews whenever possible and apologize to the authors of the numerous original papers that were not explicitly cited. By applying lipidomics approaches to knockout or transgenic mice for various PLA_2_s, it has become evident that individual enzymes regulate specific forms of lipid metabolism, perturbation of which can be eventually linked to distinct pathophysiological outcomes. Knowledge of individual PLA_2_s acquired from studies using animal models is now being translated to humans. Note that the designation “PLA_2_” is used more broadly for enzymes that have significant homology with prototypic PLA_2_ subtypes, even if they do not necessarily exert PLA_2_ activity. These examples include, for instance, *N*-acyltransferase (cPLA_2_ε), ω-*O*-acylceramide synthase (PNPLA1), triglyceride lipase (PNPLA2 and PNPLA3), and lysophospholipase (PNPLA6 and PNPLA7). Several PLA_2_s, such as cPLA_2_δ, iPLA_2_γ, and PLA2G16, possess PLA_1_ activity that is even superior to PLA_2_ activity, although biological importance of the *sn*-1 cleavage by these PLA_2_s is largely unclear. There are also several enzymes that possess PLA_2_ activity but are not designated as PLA_2_, as exemplified by the ABHD and GPI-specific PLA_2_ families. Thus, the term “PLA_2_” is somehow confusing and should be interpreted with caution if researchers encounter the genetic symbol “*PLA2*” on genome-wide transcriptome and proteome analyses.

Lipid metabolic pathways are complex, often redundant and highly interconnected, in some cases being counter regulatory. As recently reviewed by Kokotos and coworkers [221], various PLA_2_ subtypes are being targeted pharmacologically to alleviate the symptoms of various disease models, but none of the PLA_2_ inhibitors currently developed has reached the market yet. If the modulation of one PLA_2_ pathway does not suffice therapeutically, targeting two or more pathways could be effective. On the other hand, blunting of related PLA_2_ subtypes simultaneously may not be therapeutically efficient in some cases, since this strategy would block both positive and negative pathways regulated by distinct isoforms. For instance, pan-sPLA_2_ inhibitors (e.g., varespladib), which inhibit group I/II/V/X sPLA_2_s altogether, have been reported to exert a poor therapeutic effect on atherosclerosis, arthritis, and allergy, probably because they blunt both offensive and defensive sPLA_2_s. This points to potential prophylactic or therapeutic use of an agent that could specifically inhibit a particular PLA_2_ subtype, even though such a strategy will be a challenge since individual PLA_2_s appear to play highly selective and often opposite roles in specific organs and disease states.

At present, the tools that are available for following the dynamics of individual PLA_2_s and associated lipid metabolites and for monitoring their precise modifications and spatiotemporal localizations are still technically limited. The actions of lipids are frequently masked by the large steady-state mass of structural lipids in membranes, making it difficult to detect spatiotemporal lipid dynamics and functions. Further advances in this field and their integration for therapeutic use are likely to benefit from improved, time- and space-resolved lipidomics technology for monitoring individual PLA_2_s and associated lipid metabolisms within tissue microenvironments. It seems that more work will be necessary to dissect each of the many regulated pathways of bioactive lipids and to define the mechanisms responsible for the regulatory actions of individual PLA_2_s, as well as the roles of the pathways in specific responses at the cell, tissue, and organism level. Hopefully, the next decade will yield a more integrated view of the overall map of the PLA_2_ network in biology, allowing the therapeutic application of inhibitors or the lipid products of some enzymes to human diseases.

## Figures and Tables

**Figure 1 biomolecules-10-01457-f001:**
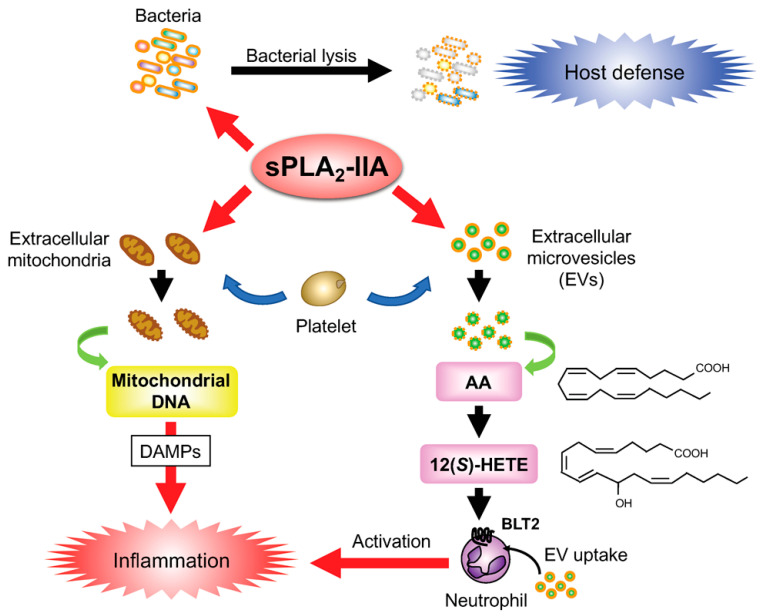
The roles of sPLA_2_-IIA in anti-bacterial defense by degrading bacterial membrane and in sterile inflammation by releasing pro-inflammatory eicosanoids from extracellular microvesicles (EVs) derived from inflammatory cells.

**Figure 2 biomolecules-10-01457-f002:**
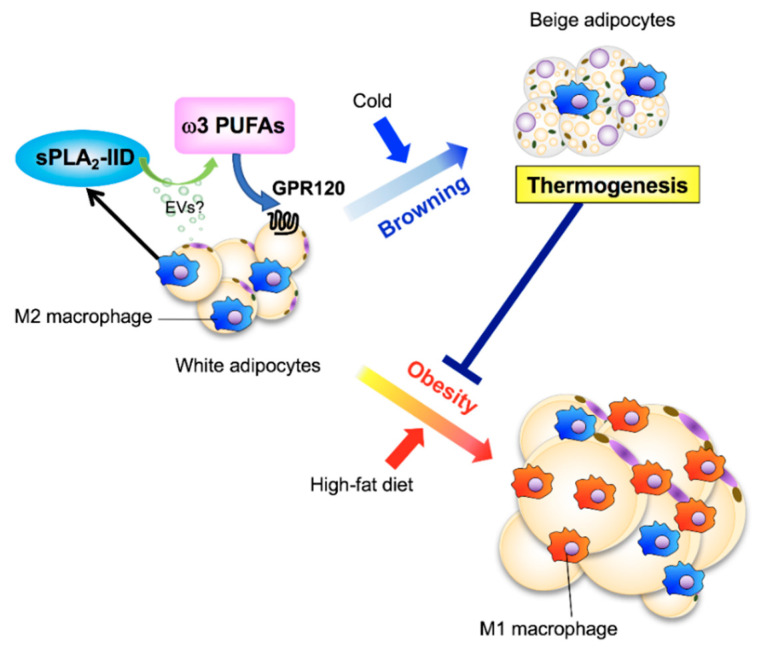
The role of sPLA_2_-IID expressed in M2 macrophages in adipocyte browning and adaptive thermogenesis. sPLA_2_-IID releases ω3 PUFAs, which then act on GPR120 to drive the thermogenic and anti-inflammatory programs toward metabolic health. Impairment of this sPLA_2_-IID-driven lipid pathway leads to impaired thermogenesis and exacerbated diet-induced obesity.

**Figure 3 biomolecules-10-01457-f003:**
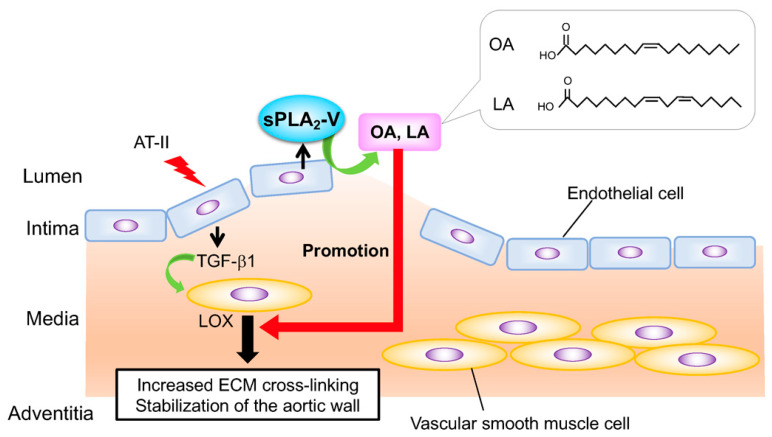
The role of endothelial sPLA_2_-V in aortic stability. sPLA_2_-V is a major sPLA_2_ isoform expressed in aortic endothelial cells (ECs) and is largely retained on the luminal surface of the aortic endothelium likely through binding to heparin sulfate proteoglycans. Endothelial sPLA_2_-V acts on membrane phospholipids of AT-II-activated ECs to mobilize oleic acid (OA) and linoleic acid (LA), which in turn promote AT-II-induced upregulation of lysyl oxidase (LOX) that facilitates ECM crosslinking, thereby stabilizing the aortic wall. Impairment of this sPLA_2_-V-driven lipid pathway leads to increased susceptibility to aortic dissection.

**Figure 4 biomolecules-10-01457-f004:**
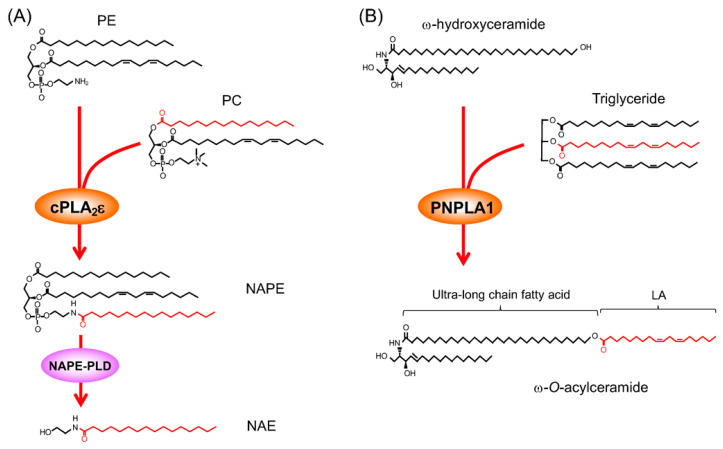
Examples of PLA_2_ enzymes that have unique enzymatic activity. (**A**) cPLA_2_ε catalyzes an *N*-acyltransferase reaction, transferring the *sn*-1 fatty acid of PC to the amino group of PE to produce NAPE, which is then converted to NAE by NAPE-PLD. (**B**) PNPLA1 acts as a unique transacylase, transferring LA from triglyceride to ω-hydroxyceramide to give rise to ω-*O*-acylceramide, which is essential for skin barrier function.

**Figure 5 biomolecules-10-01457-f005:**
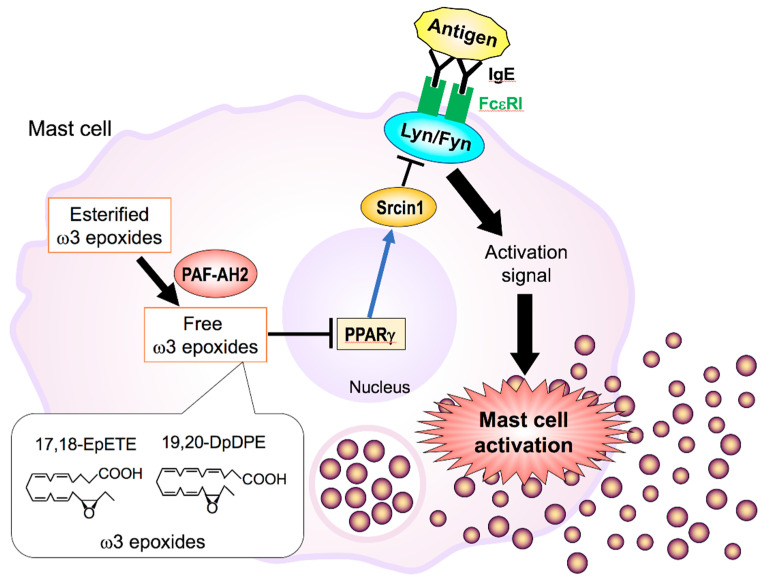
The role of PAF-AH2 in mast cell activation by producing ω3 epoxides. PAF-AH2 constitutively hydrolyzes ω3 epoxide-esterified phospholipids in cell membranes to liberate ω3 epoxides. These unique ω3 PUFA metabolites attenuate PPARγ signaling and downregulate Srcin1, which blocks activation of the Src family kinases Fyn and Lyn, thereby augmenting FcεRI signaling.

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
