# Peer review of "Updating Phospholipase A2 Biology"

_biomolecules, 2020, doi:10.3390/biom10101457_

Round 1
Reviewer 1 Report
Dear Editor, I have read with interest the manuscript provided by Dr. Murakami and collaborators on Phospholipase A2.
In this extensive and very well-written review, the Authors discuss several roles of PLA2 in the most diverse physiological process, both in health and sick conditions.
Actually I appreciated the rigorous classification of studies done by the Authors, and the value of their extensive survey. Surely, it will be appreciated by specialists in the field as well as by newcomers that have to evaluate possible directions of their research.
Honestly, there are not major issues to refer to. Authors prepared an excellent text that can be accepted essentially as it is.
Here some very minor points
- 1) The review covers “only” the past 5 years, and contains 216 references. The need of being focused and not write a “mega” review is indeed positive. However, some readers would take advantage of the great expertise of the Authors and ask for reviews that cover older literature. My suggestion is to add a short Appendix, where the Authors can cite some key reviews published in the past, and that can be used by readers as starting point for further readings. For example, the Authors could explicitly mention that their review refers only to the past 5 years, whereas readers interested in older reviews should refer to X, Y and Z because X is dedicate to …, Y focuses on …. , etc (i.e. brief comments of the suggested reviews). I believe this will be an important addition to the current study.
- 2) A second question is about PLA2 operating on lipids of opposite stereochemistry. Maybe this is not of primary relevance for human physiology but it is related to progresses in the field. What about PLA2 operating on sn-1 phospholipids? Any relevant news?
- 3) Please be sure that N- and C- (in N-terminal or C-terminal sentences) are in italics everywhere.
- 4) In the concluding remarks the Authors have partially expressed their opinion about the field and future directions. Maybe the readers would appreciate a slightly more extended discussion here.
Author Response
Dear Reviewer 1,
Thank you very much for your very positive comments. According to your suggestions, we have made following minor revisions.
1) The review covers “only” the past 5 years, and contains 216 references. The need of being focused and not write a “mega” review is indeed positive. However, some readers would take advantage of the great expertise of the Authors and ask for reviews that cover older literature. My suggestion is to add a short Appendix, where the Authors can cite some key reviews published in the past, and that can be used by readers as starting point for further readings. For example, the Authors could explicitly mention that their review refers only to the past 5 years, whereas readers interested in older reviews should refer to X, Y and Z because X is dedicate to …, Y focuses on …. , etc (i.e. brief comments of the suggested reviews). I believe this will be an important addition to the current study.
Answer: We have added the following sentence to the last part of the Introduction.
“Readers interested in older views as a starting point for further readings should refer to our current reviews describing the classification of the PLA2superfamily [1], those covering PLA2s and lipid mediators broadly[2,3], and those focusing on sPLA2s [4-7].”
2) A second question is about PLA2 operating on lipids of opposite stereochemistry. Maybe this is not of primary relevance for human physiology but it is related to progresses in the field. What about PLA2 operating on sn-1 phospholipids? Any relevant news?
Answer: Sorry, we have no idea on PLA2or PLA1 operating on lipids of opposite stereochemistry, since we do not think it relevant to physiology.If the reviewer would know it, please tell us how we should add it.
3) Please be sure that N- and C- (in N-terminal or C-terminal sentences) are in italics everywhere.
Answer: We have changed them to Italics. Thank you.
4) In the concluding remarks the Authors have partially expressed their opinion about the field and future directions. Maybe the readers would appreciate a slightly more extended discussion here.
Answer: We have extended the Discussion.
Reviewer 2 Report
This work presents an important and large review of classification and properties of lipolytic enzymes belonging to the various families of phospholipase A2 and the author makes here a catalogue of the different and numerous functions and beneficial effects of different families of phospholipase A2. The authors remarks that they focus on findings in last five years. There are very many papers and reviews on this subject including the reviews of the authors published previously. Many aspects have already been covered in the literature and published reviews. So, for example, information on the classification of this enzyme briefly as a table can be found in the previous publication of Dr. Murakami (2019). This is why, in my opinion, it would be useful to focus to what new knowledge we can get from this just another review.
In my opinion, the greatest problem is duplication of text from previously published papers that I consider unacceptable.
While in review I have been noticed that at least part of this submission appears to have been published already. So, Abstract and at least the beginning of the Introduction were copied from the paper “Novel functions of phospholipase A2s: Overview.” BBA - Molecular and Cell Biology of Lipids 1864 (2019) 763–765. https://doi.org/10.1016/j.bbalip.2019.02.005
The author should explain it and only after that reviewing of the article may be continued.
Author Response
Dear Reviewer 2,
We apologize that the introduction and abstract in the original version was largely overlapped with my previous paper “Novel functions of phospholipase A2s: Overview.” BBA - Molecular and Cell Biology of Lipids 1864 (2019) 763–765. This BBA paper was the one I myself wrote as a preface of the Special Issue, in which I worked as a guest editor.I thought that this overview (preface) is suitable for the abstract and introduction of the present review.
In response to the reviewer, we have edited the abstract and introduction considerably. Also, we have made minor revisions throughout the text.
We do hope that these changes are appropriate.
Round 2
Reviewer 2 Report
In response to my comments, the authors revised the submission and rephrased Abstract and Introduction.